# Competition Restricts the Growth, Development, and Propagation of *Carpinus tientaiensis*: A Rare and Endangered Species in China

**Liangjin Yao** [1,2], **Yuanke Xu** [3], **Bo Jiang** [1], **Chuping Wu** [1], **Weigao Yuan** [1], **Jinru Zhu** [1], **Tingting Li** [1] and **Zhigao Wang** [1,*]

1   Zhejiang Forestry Academy, Hangzhou 310023, China; lj890caf@163.com (L.Y.); jiangbof@126.com (B.J.); wcp1117@hotmail.com (C.W.); zfaywg@126.com (W.Y.); chinazjzjr@126.com (J.Z.); tingtingli@163.com (T.L.)
2   Key Laboratory of Forest Ecology and Environment of the State Forestry and Grassland Administration, Research Institute of Forest Ecology, Environment, and Protection, Chinese Academy of Forestry, Beijing 100091, China
3   Center of Ecological Forestry Development, Jingning Nationality Autonomous County, Lishui 323500, China; jnxyk026@126.com
*   Correspondence: wangzg78@126.com

**Abstract:** The protection and propagation of rare and endangered species are key to the preservation of their population development; however, due to the scarcity of individuals, the potential effects and status of rare and endangered species in the whole forest ecosystem are still poorly understood. Using data from a 60 × 140 m forest dynamic monitoring sample of the *Carpinus tientaiensis* (*Betulaceae*) species in Zhejiang of Southeast China. We assessed the population distribution and diameter at breast height (DBH) structure of the *Carpinus tientaiensis* species, which was a rare and endangered species, as well as intra- and interspecific correlation with other species. The results show that saplings (1 cm ≤ DBH < 5 cm) and juveniles (5 cm ≤ DBH < 10 cm) were more aggregated than larger individuals (DBH ≥ 20 cm) of *Carpinus tientaiensis*. The DBH size structure of all the trees shows an obvious inverted "J" distribution. With an increase in the DBH size category, the number of individuals gradually decreases. Due to the diffusion limitation, the spatial distribution patterns of all the tree individuals and roof geese in the sample land are increased at a small spatial scale, and as the spatial scale increases, the degree of aggregation decreases gradually. The relationship between different diameter stages of the population of *Carpinus tientaiensis* showed a consistent general trend. The spatial distribution of individuals with a large diameter on a small scale was significantly positively correlated ($p < 0.001$). With an increase in the scale, there was no significant correlation ($p > 0.05$) between individuals with a large diameter and individuals with a small diameter. There was no significant correlation ($p > 0.05$) between the population of *Carpinus tientaiensis* and other species in the sample, and the strong unidirectional competition of other species in the sample can be seen by the competition index. We found that interspecific competition restricts the growth and expansion of *Carpinus tientaiensis*, and it has adopted different ecological strategies to coexist with a population of common tree species occupying a similar living space.

**Keywords:** spatial correlation; ecological strategy; intra- and interspecific interactions; conservation of rare species; *Carpinus tientaiensis*

## 1. Introduction

Exploring the coexistence mechanism of species in natural forest communities is one of the core aspects of community ecology [1,2]. Intra- and interspecific interactions (competition and facilitation) have a significant role in constructing community structures and maintaining a stable community coexistence [2–7]. Competition not only causes the death and self-preservation process caused by a density limitation of the population [7,8], but also

leads to changes in the relationship between the plastic growth and the allometric growth of the competition among the individual species [9]. Competition to plunder each other's living space and resources to affect the survival and growth of species causes changes in the community's spatial structure and species composition [10,11]. The environmental heterogeneity is different on a larger scale, while the survival rate and diameter level of the same species are different in different habitats. The individuals are affected by a combination of habitat association and diffusion restriction, and competition among tree species may be asymmetric [12,13]. Some studies suggest that the difference in tree diameter level intensifies the asymmetric competition of resources and affects the community structure and spatial pattern of trees [2,14]. Individuals with a large diameter have a stronger competitive ability and ability to resist and adapt to harsh environments, which can inhibit the survival of individuals with a small diameter through shading and allelodiametery [15].

Plant competition is a neighborhood phenomenon, and the spatial distribution of species plays an important role in competition. The distribution pattern of plants will determine the range of species competition, and the size of adjacent individuals is also one of the factors that determine the intensity of competition [16]. Based on the spatial distribution of species, we gained insight into the potential ecological processes, plant growth, diffusion, death, and other processes related to the degree of spatial distribution aggregation [11,12,17]. Similarly, such approaches have helped us to explain the promotion mechanisms of multi-species coexistence and predict the successional processes and structural dynamics of communities [18]. The spatial distribution pattern of species is an important basis for plant population characteristics, population interaction, and the relationship between populations and the environment [19,20]. Changes in species, tree diameters, and spatial scales can lead to changes in the degree of aggregation (aggregation, random, and uniform distribution) of tree species [10,17]. The analysis of spatial correlations among species (e.g., between trees of different diameter sizes or between different tree species) may reflect the results of species competition for limited resources [8] or reveal the species utilization of living resources and adaptation to habitats [21]. By studying the spatial patterns and connections of three dominant species on Vancouver Island, Canada, Getzin et al. [16] employed the theory of intraspecific competition and interspecific competition to maintain community structure. Similarly, Martinez et al. [22] found that seed diffusion, interspecific interactions, and external disturbances all affected the spatial distribution patterns in temperate forests in Northwest Spain. Liu et al. [2] found that competition and promotion affected the spatial pattern of plants in the cold temperate forest of Kanas, Xinjiang, China, and the effect of promotion on the survival and growth of trees was more intense. Additionally, species with different life-history strategies also have different spatial distribution patterns [23]. For example, light-preferring species usually exhibit aggregation distribution at small diameter, while showing random distribution at larger diameter, which might be due to increased resource competition. Therefore, according to the spatial distribution pattern of species, it is possible to speculate on intraspecific and interspecific interactions and analyze the correlation between species and habitats [18]. Exploring the spatial correlation between different populations of communities can provide information on many potential ecological processes for exploring forest patterns and processes and better understanding the mechanism of population maintenance and dynamic change [10,11,24].

*Carpinus tientaiensis* Cheng is a deciduous tree of the genus Betulaceae, which is a rare and endangered species unique to China and an ancient residual species of the Upper Eocene in the Tertiary [25]. It is listed on the Red List of critically endangered species of the International Union for the Conservation of Nature (IUCN, [26]), and a similar species, *Carpinus tientaiensis*, as critically endangered [26]. Its low pollen vigor and incomplete seed embryo development lead to a low seed setting rate and high seed shell rate, which are the main reasons for the decrease in the population of *Carpinus tientaiensis*. In forests >900 m above sea level in Huading National Forest Park in Tiantai Mountain (29°09′–29°28′ N, 120°50′–121°24′ E), only 19 wild plants were found previously. In this study,

we found 351 *Carpinus tientaiensis* plants growing on both sides of the gully and in the evergreen broad-leaved forest on the Shangtoushan in Jingning, Zhejiang Province. This is the largest population found at present, and it is helpful for broadening the protection of this population and understanding the habitat adaptability of *Carpinus tientaiensis* [25]. Studying the spatial structure and competitiveness of *Carpinus tientaiensis* populations is helpful in species protection, as it aids in the classification of the flora of *Betulaceae* and endangered populations. To understand the spatial pattern of *Carpinus tientaiensis* and the mechanism that affects species coexistence in their populations, we used scale-dependent point pattern analysis to quantify the spatial distribution and association patterns of different diameter-size sequences of species from different populations. We used the following questions to guide the analyses: (1) What is the distribution pattern of *Carpinus tientaiensis* populations? (2) Does the *Carpinus tientaiensis*-based spatial association vary with the DBH size of trees, and is its facilitation more frequent than its competition? (3) What are the reasons for restricting *Carpinus tientaiensis* population expansion in a relatively harsh *Carpinus tientaiensis* community environment?

## 2. Materials and Methods

### 2.1. Study Area

Jingning Shangtoushan (27°31′–27°56′ N, 119°30′–121°55′ E) is located 21 km south of the county seat of She Autonomous County, Jingning, Zhejiang Province, located at the junction of Daji Township and Jingnan Township. This area has many low hills and mountain terraces, and it is the main peak of the right branch of the Donggong Mountains. It has a subtropical monsoon with a warm, humid climate, abundant rainfall, and four distinct seasons. The highest altitude is 1689.1 m asl (above sea level); the annual average rainfall is 1918 mm; the annual average temperature is 11.8 °C; the coldest monthly average temperature is 2.1 °C; the hottest monthly average temperature is 21.5 °C; the average relative humidity is more than 80%; the annual sunshine time is 1717.6 h; and the frost-free period is about 196 day. The daily average temperature is $\geq$10 °C, and the annual accumulated temperature is 2858–5157 °C. The annual growth period of vegetation is greater than 200–240 day. As a result of the complex terrain, the altitude is different and the climate vertical difference is obvious. The forest coverage rate is 96.5%, and the main tree species are *Rhododendron fortune*, *Symplocos coreana*, *Pinus taiwanensis* [25] (Table 1).

**Table 1.** Abundance, basal area (BA), and competition index shade tolerance of the common species in the *Carpinus tientaiensis* sampling plot. Shade tolerance categories for each species were obtained from Lu and Yan [27].

| Species | Family | Abundance (Tree·ha$^{-1}$) | Basal Area (m²·ha) | Competition Index ($CI_1$) | Shade Tolerance |
|---|---|---|---|---|---|
| *Rhododendron fortunei* | *Ericaceae* | 1243 | 7.74 | 0.316 | Shade-tolerant |
| *Symplocos coreana* | *Symplocaceae* | 448 | 1.93 | 0.212 | Light-demanding |
| *Carpinus tientaiensis* | *Betulaceae* | 351 | 4.9 | 0.203 | Light-demanding |
| *Eurya hebeclados* | *Theaceae* | 255 | 1.28 | 0.264 | Shade-tolerant |
| *Lyonia ovalifolia* | *Ericaceae* | 145 | 0.52 | 0.188 | Light-demanding |
| *Pinus taiwanensis* | *Pinaceae* | 120 | 3.98 | 0.227 | Shade-tolerant |
| *Cyclobalanopsis multinervis* | *Fagaceae* | 80 | 1.35 | 0.295 | Shade-tolerant |

### 2.2. Sample Surveys and Data Collection

From July to August 2019, a 60 × 140 m forest dynamic monitoring sample of the *Carpinus tientaiensis* species was established in Shangshantou, Jingning, Zhejiang Province, according to the standard for sample construction of the Center for Tropical Forest Science. The whole plot was divided into 21 20 × 20 m samples by the total station, and then all of the 20 × 20 m samples were divided into 16 small quadrats of 5 × 5 m. The surviving woody plants, with a DBH $\geq$ 1 cm in each sample, were numbered with an aluminum plate and marked with red paint at a height of 1.3 m. The species names, DBH, tree height,

coordinates, branching, and germination status of all labeled woody plant individuals were recorded.

According to the classification criteria of the diameter-level structure of canopy trees in subtropical and temperate zones [2,28], the characteristics of the distribution about the radial order of the population of *Carpinus tientaiensis* were combined. We divided them into four categories according to their DBH: saplings as individuals with a 1 cm $\leq$ DHB < 5 cm; juveniles with a 5 cm $\leq$ DHB < 10 cm; middle trees with a 10 cm $\leq$ DHB < 20 cm; adults with a DBH $\geq$ 20 cm.

### 2.3. Competition Index

The competition index is a useful index with which to study competition among species. It reflects the adaptive relationship between the growth of individual trees and their living space and essentially reflects the occupation of environmental resources by trees in real habitats. Using the distance-related index proposed by Hegyi [29], we can effectively reflect the utilization degree of environmental resources by plant individuals. The calculation method is as follows:

$$CI_1 = \sum_{j=1}^{N} D_j D_i^{-1} L_{ij}^{-1} \tag{1}$$

where $i$ is the object wood, $j$ represents the individuals competing against object wood, $CI_1$ is the competition index of the object wood $i$, $D_j$ is the DBH of the individuals which competing against object wood $j$, $D_i$ is the DBH of the object wood $i$, $L_{ij}$ is the distance of the object wood $i$ to the individuals which competing against object wood $j$, $N$ is the number of competing trees around the object wood $i$. A larger $CI_1$ value represents a more intense competition of the object wood.

### 2.4. Point Pattern Analysis Method and Null Model

The pair-correlation function g(r) was used to analyze the spatial distribution of individual and population levels. The pair-correlation function g(r) is derived from the K function, mainly using the circle to replace the circle in the K function, excluding the cumulative effect in the calculation process [18]. That is, the degree of the spatial aggregation of species is analyzed by the single variable g(r) function, and the spatial correlation between two species is analyzed by the double variable g(r) function.

Null model selection: Environmental heterogeneity affects species distribution at large scales, and competition among species may be asymmetric [12]. Considering that the survival rate of the same species is different in different habitats, individuals are jointly affected by habitat association and diffusion limitation. Therefore, the heterogeneous Poisson process is used to simulate the single species and bivariate statistics. The Heterogeneous Poisson (HP) process is mainly used to simulate the relationship between the species density function and its habitat in order to exclude the zero hypothesis model of habitat heterogeneity on a large scale and predict the probability of the occurrence of a population individual in a region with environmental covariates as a function. λ(s) shows the relationship between individual density and habitat heterogeneity through the spatial heterogeneity function [2,30]. In the simulation process, we first fixed the position of one tree species, randomized the spatial distribution position of another tree species by the Heterogeneous Poisson process, and analyzed the changes of the spatial correlation between the two tree species. Then, keeping the position of the latter tree species unchanged, the spatial distribution of the former tree species was randomized by the Heterogeneous Poisson process, and the changes of the spatial correlation between the two tree species were analyzed again. A confidence interval of 99% was obtained using 100 Monte Carlo simulations, and the maximum distance scale was 60 m [18]. Concerning the univariate analysis, if the observed value was above the confidence interval, this indicated aggregation; if the observed value was below the confidence interval, this indicated a random distribution; and if the observed value was between the confidence intervals, this indicated

a regular/hyperdispersed pattern. For bivariate analysis, if the observed value was above the confidence interval, this indicated a positive correlation; if the observed value was below the confidence interval, this indicated a negative correlation; and if the observed value was between the confidence intervals, this indicated that there is no significant correlation between the two species [18]. All the analyses were conducted using the "spatstat" package in R3.2.4 [31].

## 3. Results

### 3.1. Individual Structure

A total of 3343 individuals were investigated in *Carpinus tientaiensis* plots. Among them, we analyzed the seven common species, which accounted for 76.6% of the total relative abundance in the sample. *Rhododendron fortunei* was the most common species, accounting for 37.2% of the total abundance, and *Symplocos coreana* and *Carpinus tientaiensis* accounted for 13.4% and 10.5% of the relative abundance, respectively. The relative abundances of the other four species (*Eurya hebeclados*, *Lyonia ovalifolia*, *Pinus taiwanensis*, and *Cyclobalanopsis multinervis*) are small; however, *Rhododendron fortunei* had the highest relative basal area (28.1%), followed by *Carpinus tientaiensis* (17.8%) and *Pinus taiwanensis* (14.4%). The DBH size structure of all trees and *Carpinus tientaiensis* in the sample land is similar, the diameter structure of each population showed a characteristic reverse J distribution, and the number of individuals decreased with the increasing diameter size in the sample (Figure 1). A large number of the individuals around the *Carpinus tientaiensis* with a DBH of between 1 and 15 cm are saplings and young trees (Figure 1).

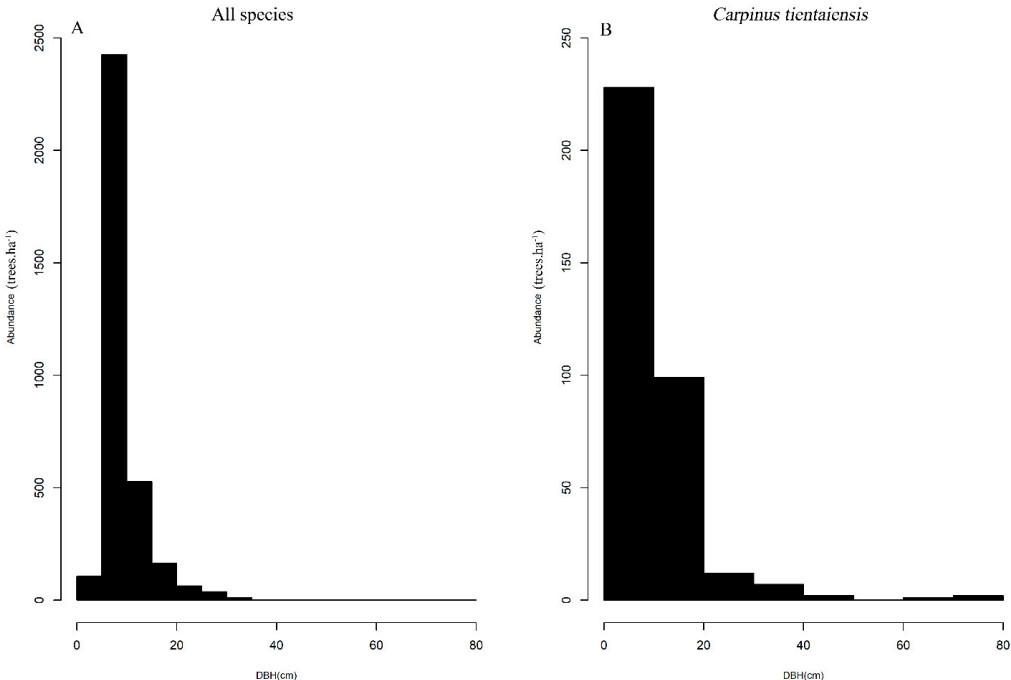

**Figure 1.** Diameter at breast height (DBH) structure of all species (**A**) and *Carpinus tientaiensis* (**B**).

### 3.2. Spatial Distribution Patterns of Carpinus tientaiensis and Other Species

Overall, all the species in the plots exhibited an aggregation distribution at a scale of 0–5 m, a random distribution at a scale of 5–18 m, and a uniform distribution at a larger scale (r ≥ 18 m). All individuals of *Carpinus tientaiensis* exhibit an aggregation distribution at a scale of 0–13 m, but they have a random distribution in region m of r ≥ 13 (Figure 2). The saplings, juveniles, middle trees, and adults of the *Carpinus tientaiensis* population all showed a small-scale aggregation distribution (r ≤ 5 m) and a large-scale random distribution (5 m < r). With the increase in DBH, the aggregation degree decreased to a random distribution (Figure 3A–D).

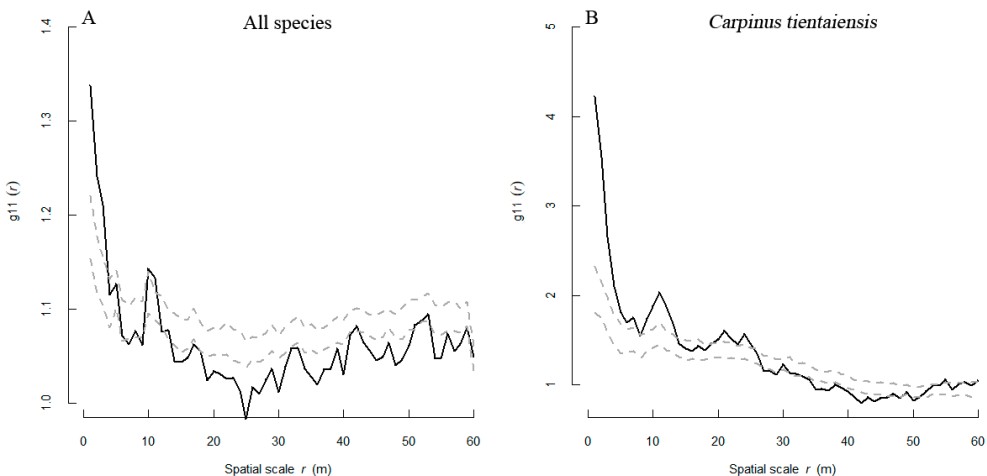

**Figure 2.** Spatial patterns of all species (**A**) and *Carpinus tientaiensis* (**B**) in the *Carpinus tientaiensis* sampling plot. Note: Black lines indicate the $g_{11}(r)$ function, dotted lines indicate the upper and lower limits of the 99% confidence interval. Observed values above the upper limits indicate aggregation, within the intervals indicate random distributions, and below the lower limits indicate a regular/hyperdispersed pattern.

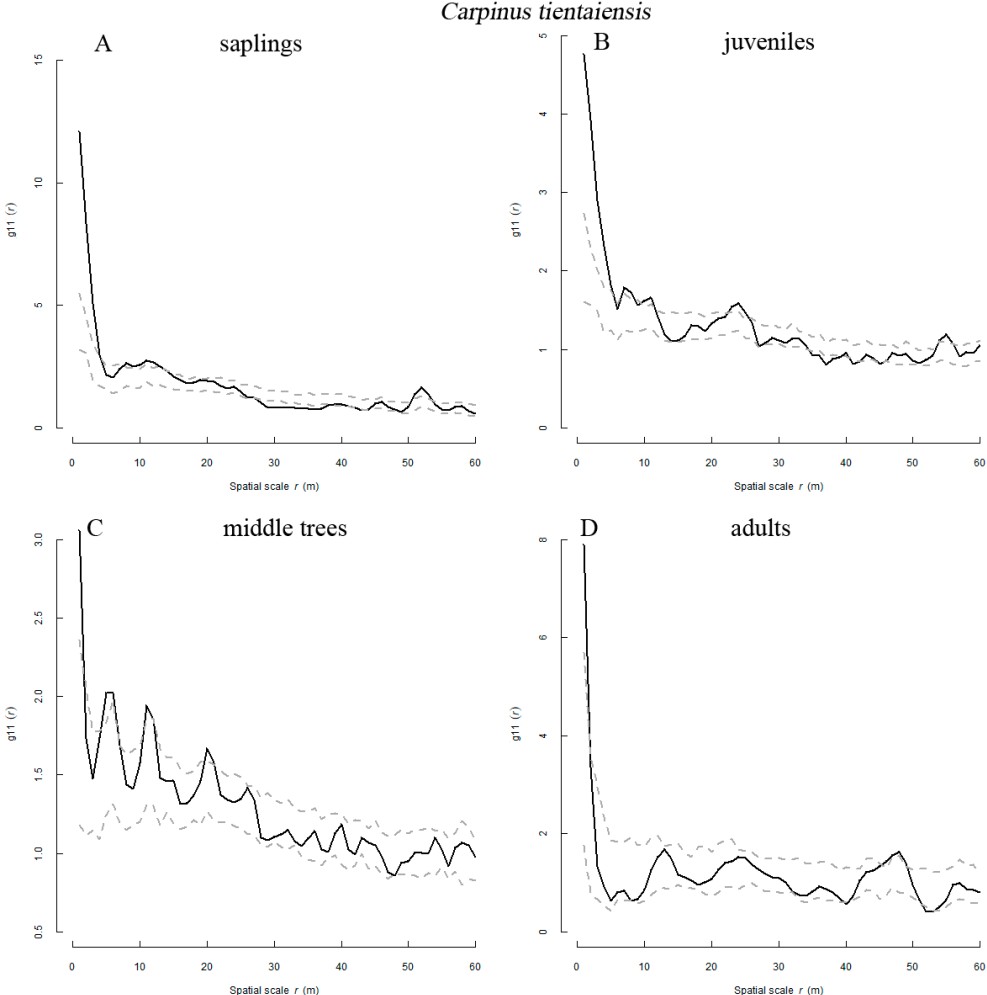

**Figure 3.** Spatial patterns of *Carpinus tientaiensis* among different DBH classes in the *Carpinus tientaiensis* sampling plot. Note: The insertions at figure a indicate the point patterns analysis of three size classes (saplings as 1 cm $\leq$ DBH < 5 cm, juveniles as 5 cm $\leq$ DBH < 10 cm, middle trees as 10cm $\leq$ DHB < 20 cm and adults as DBH $\geq$ 15 cm).

### 3.3. Intraspecific Association of Carpinus tientaiensis Population

The relationship between different diameter levels of the *Carpinus tientaiensis* population shows a consistent overall trend. There was a significantly positive correlation ($p < 0.001$) between individuals with a large diameter and individuals with a small diameter on a small scale. With an increase in the scale, there is no significant correlation between individuals with a large diameter and individuals with a small diameter (Figure 4A–F).

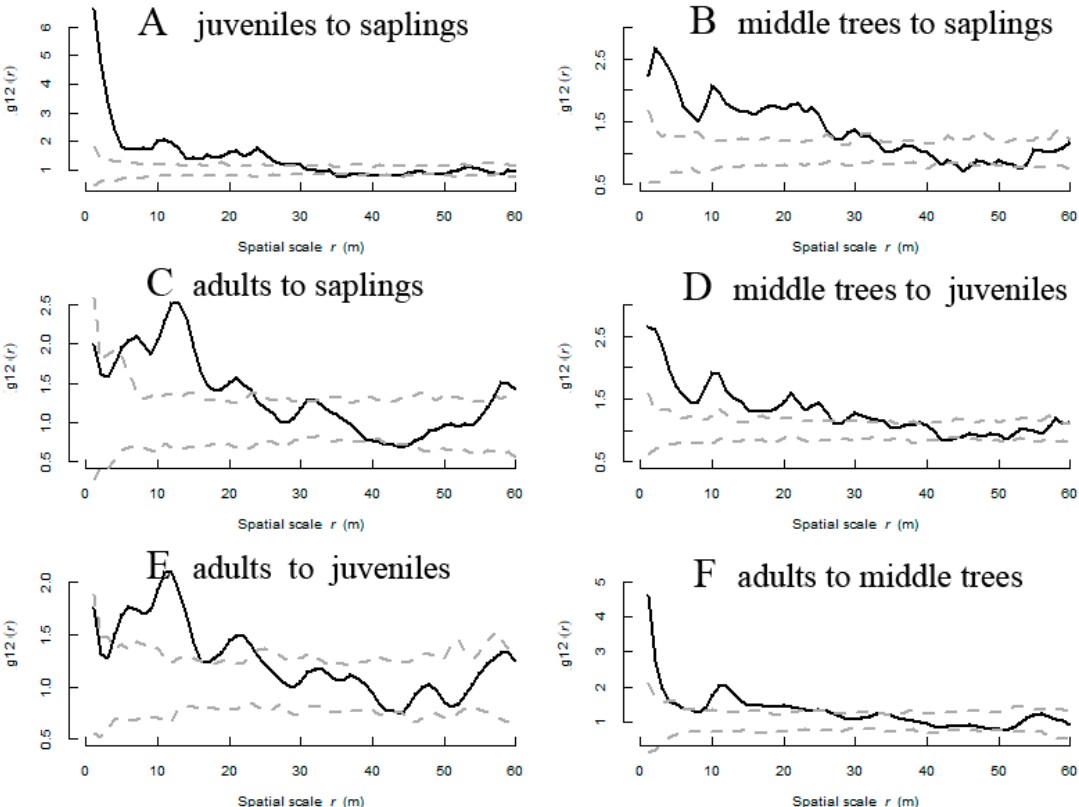

**Figure 4.** Spatial associations of *Carpinus tientaiensis* among different DBH classes in the *Carpinus tientaiensis* sampling plot. Note: The insertions at figure a indicate the point patterns analysis of three size classes (saplings as 1 cm $\leq$ DBH < 5 cm, juveniles as 5 cm $\leq$ DBH < 10 cm, middle trees as 10 cm $\leq$ DHB < 20 cm and adults as DBH $\geq$ 15 cm). Black lines indicate $g_{12}(r)$ function, dotted lines indicate the upper and lower limits of the 99% confidence interval. Observed value above the upper limits indicate positive association, within the intervals indicate no association, and below the lower limits indicate negative association.

### 3.4. Interspecific Association of Carpinus tientaiensis Population

Individuals of different diameter levels of *Carpinus tientaiensis* populations are affected differently by other species in the sample. Other species with a small diameter level in the sample showed a significantly positive correlation to *Carpinus tientaiensis* saplings on a small scale ($r \leq 30$ m), but with an increase in the spatial scale, the correlation gradually weakened (Figure 5A,B).

Other species with a medium diameter level (middle trees) in the sample plot showed no significant correlation to the saplings, juveniles, and middle trees of the *Carpinus tientaiensis* population at any scale (Figure 5C–E). Other species with a large diameter (adults) in the sample showed a significantly positive correlation to seedlings of the *Carpinus tientaiensis* population on a scale of 30–40 and no significant correlation on the other scales (Figure 5F). Other species with a large diameter (adults) in the sample showed no significant correlation to the juveniles, middle trees, and adults of the *Carpinus tientaiensis* population at any scale (Figure 5G–I).

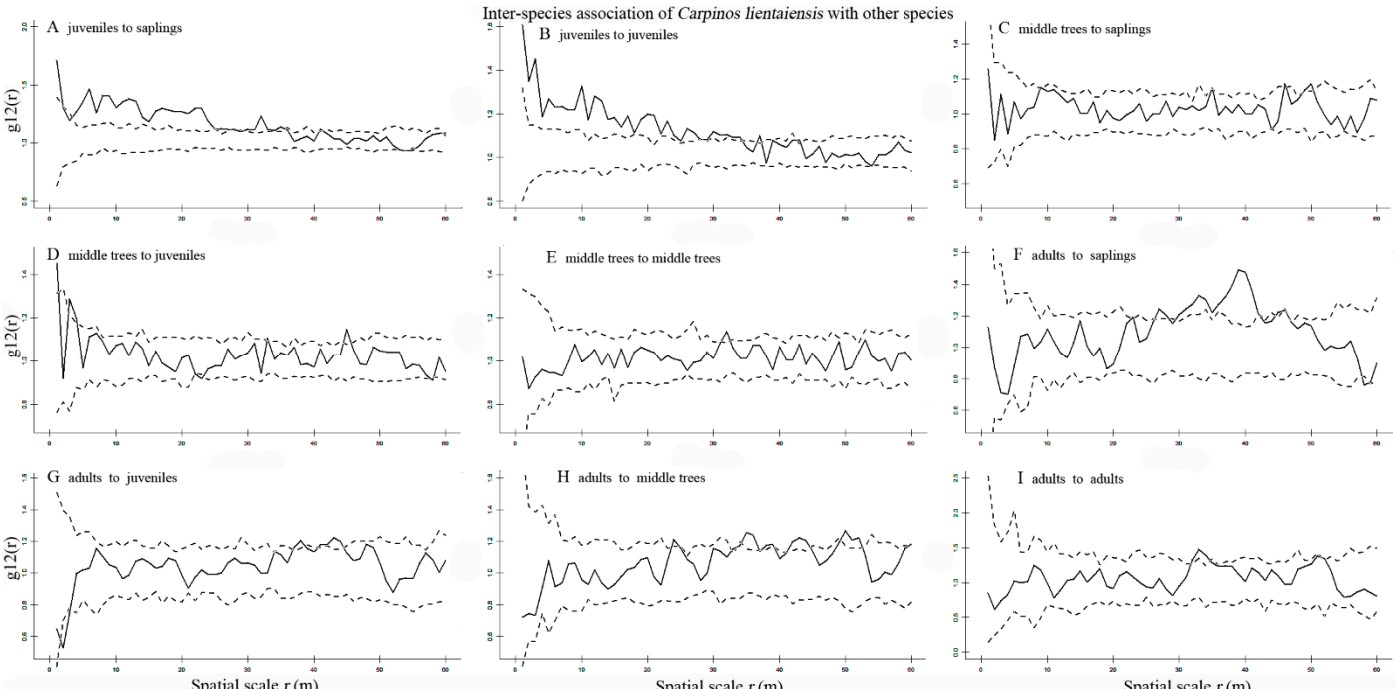

**Figure 5.** Spatial associations of *Carpinus tientaiensis* to other trees among different DBH classes in the *Carpinus tientaiensis* sampling plot. Note: The explanation for all curves and describe each subfigure individually as Figure 4.

Based on the competition index of the main common species in the sample, the competition index of *Carpinus tientaiensis* populations (0.203) was relatively small among the species in the plots and much lower than, for example, that of *Rhododendron fortune* (0.316), *Cyclobalanopsis multinervis* (0.295), *Eurya hebeclados* (0.264), etc. It was only higher than that of *Lyonia ovalifolia* (0.188).

## 4. Discussion

Our research supports the view that competitive action is more common and critical in the process of the colonization, growth, and development of rare and endangered plants. Among the *Carpinus tientaiensis* communities, the competitive ability of shade-tolerant tree species is relatively strong, while that of strong light-preferring tree species is relatively weak. While the weaker promotive effect of different size grades is observed within and between *Carpinus tientaiensis* species. However, the frequency and intensity of promotion was much lower than that of competition due to habitat or other conditions. Therefore, weak promotion may not be the main driving force of community construction. We speculate that the negative effect of the *Carpinus tientaiensis* population (i.e., competition) plays a more important role in the growth and development of the *Carpinus tientaiensis* population in this area.

### 4.1. Intraspecies Interactions of Carpinus tientaiensis Populations

All species and individuals of different diameter levels in the *Carpinus tientaiensis* population show an aggregation distribution on a small scale. Usually, the degree of aggregation decreases with an increase in the spatial scale, which is consistent with the observation in subtropical and temperate forest communities, and accords with the results of most studies on population spatial patterns [2,24,32]. The aggregation distribution of plants is common in natural communities. Early studies have shown that species populations are clustered at a certain scale, both in tropical and temperate zones [33]. It was determined that, in the tropics, the degree of aggregation did not decrease significantly with an increase in the scale; however, in temperate forests, the species change from aggregation distribution to random distribution with an increase in the scale [2,34]. Some

studies have suggested that, in a small number of species, the aggregation trend is often higher, and the larger the diameter, the more random the distribution [1,10]. As a result, the *Carpinus tientaiensis* individuals with a large diameter had the lowest number, and the degree of aggregation is the lowest. Our results are in agreement with previous conclusions, suggesting that, in *Carpinus tientaiensis* plots, the species spatial patterns on a large scale are mainly affected by species' own habitat preferences, while the intraspecific aggregation on a smaller scale is driven by the size of the tree diameter, the studied scale, and the relative abundance of the species [16,24,32,35].

The interaction between individual trees in the community was closely related to the spatial scale and diameter scale [36]. We found that the spatial distribution of individuals of the *Carpinus tientaiensis* population with a small diameter level on a small scale is significantly positively correlated, and with an increase in the scale, there was no significant correlation between individuals of a large diameter and those of a small diameter. This suggests that most small individuals are distributed around their parents due to dispersion restrictions. Pacala [37] believed that the dispersion restriction leads to the aggregation of intraspecific patterns and the pattern of interspecific separation. Individuals of the *Carpinus tientaiensis* population are clustered in small scales but are random on all other scales. When the individual diameter level increases, as the intraspecific competition intensifies, the degree of aggregation around individual trees will decrease significantly [10,38]. Competition among individuals, the spatial distribution of *Carpinus tientaiensis* trees is often more dispersed than that of small-scale saplings and young trees. Again, some studies have suggested that the degree of aggregation of individuals with a large diameter was reduced due to the influence of self-sparsity caused by competition [1]. As we speculated, other species in the *Carpinus tientaiensis* community tend to be highly dispersed on smaller scales with an increase in the diameter level, which may be driven by intraspecific competition, and this is also in line with the conclusion predicted by the classical Janzen–Connell hypothesis [2,39]. This also indicates that the mortality of the population is dependent on the density, which leads to an increase in the excessive dispersion of trees from a small diameter level to a large diameter level [13,16]. Our results showed that the competition of resources among individuals with a large diameter was more intense than that of individuals with a small diameter, so the spatial pattern of individuals with a large diameter was more regular [40]. At smaller scales, intraspecific and interspecific interactions affect species distribution patterns [35]. Therefore, on a smaller spatial scale, resource competition leads to more obvious intraspecies and interspecific relationships, but their diameter differences are one of the determinants of interactions between individuals [2]. The results showed that the spatial distribution of *Carpinus tientaiensis* populations tends to be small-scale aggregation to large-scale random variation patterns. We did not find intraspecific competition in individuals of different diameter levels of the *Carpinus tientaiensis* population, and the individuals with a large diameter level promoted the growth of seedlings to a certain extent. Therefore, we found that competition intensity in the *Carpinus tientaiensis* population was very small [25].

### 4.2. Interspecific Interactions of Carpinus tientaiensis with Other Species

Interspecific competition processes affect the settlement, growth, and stable coexistence of other species within tropical, subtropical, and temperate forest communities [2,16,41]. Based on the spatial association analysis and competition index results among different species, we found that there is an intense unidirectional interspecific competition between *Carpinus tientaiensis* populations and other species during their growth and development, e.g., the competitive role of the shade-tolerant species, *Rhododendron fortunei* and *Cyclobalanopsis multinervis*, on the light-demanding species, *Carpinus tientaiensis*. This one-way competition is mainly the competition of the individual tree of the canopy for the light and resources of the small individual under the forest. This is why the *Carpinus tientaiensis* population was not competitive in the community, and the adaptability to the relationship between the whole habitat and organisms is weak, so the growth is blocked, and the pop-

ulation size is small. As a result, we speculate that asymmetric interspecific competition plays a relatively strong role in shaping the individual structure and species coexistence of the *Carpinus tientaiensis* populations in this forest community. This suggests that the competitive effect mainly regulates the spatial pattern of *Carpinus tientaiensis* populations, and the shade-tolerant species (such as *Rhododendron fortune*, *Cyclobalanopsis multinervis*, etc.) compete for the living space and resources of *Carpinus tientaiensis* populations. In high-altitude habitats with limited resources, the competition among species for resources (such as light, mineral elements, etc.) is increasingly fierce [16]. How does the competition between species allow them to stably coexist? *Carpinus tientaiensis* is a fast-growing and light-deciduous strong species with a high specific leaf area and low specific stem density. Most successional pioneer species (e.g., deciduous species) use resource acquisition strategies to achieve rapid growth and reproduction [2]. The *Rhododendron fortune* and *Cyclobalanopsis multinervis* of the main tree species in the sample land are shade tolerant, with a low specific leaf area and high specific stem density. Most species adopt a resource conservation strategy and grow slowly to improve their competitiveness [42]. Therefore, two different ecological strategies, namely, the *Carpinus tientaiensis* resource acquisition strategy (that is, the rapid acquisition of resources to maintain individual growth and development), and the resource conservation strategy of *Rhododendron fortunei* and *Cyclobalanopsis multinervis* (slow growth but strong resistance to external adverse factors), are conducive to the coexistence of the species. We speculate that even if there is strong competition among species, the competing species can coexist as long as the species competitiveness or competition is weakened by some external factors [4,17,43]. The coexistence of *Carpinus tientaiensis* with other species may be on an ecological scale, with an appropriate frequency of environmental fluctuations (such as interference, feeding, etc.) that can prevent competitive exclusion, allow for the persistence of species-rich non-equilibrium communities, and increase the likelihood of coexistence [43].

Recently, ecologists have become increasingly aware that positive interactions among species are important factors affecting species coexistence and diversity maintenance in alpine plants and play a key role in the restoration and conservation of alpine ecosystems [2,5,24,32,44]. Especially in habitat with poor conditions such as low temperature, drought or barrenness, species collaborate to promote the survival and growth of other plants [21,45]. The low temperature, high humidity, and relatively low sunlight caused the bad habitat conditions for the local vegetation, which were also affected by the high altitude [46] in the *Carpinus tientaiensis* community. We found that other species were positively correlated with *Carpinus tientaiensis* seedlings. Previous studies have shown that the positive correlation between individuals with a small diameter level may not be due to mutual promotion between species, but it could rather be influenced by adjacent spatial locations or similar habitat preferences, such as the need for light and soil nutrients [47,48]. Influenced by habitat heterogeneity, not only are there interactions among different species, but other biological processes may also produce this positively correlated spatial pattern [4,49]. We speculated that, compared with individuals with a large diameter (big trees and middle trees), the neighborhood competition between individuals with a small diameter (young trees) among different species was relatively weak. Plant-to-plant interactions may also be affected by the environment, where species-to-species linkages may increase as environmental pressures increase [45,49]. Steinbauer et al. [49] considered that, because environmental factors affect the adjacent distribution and coexistence of species, we could not clearly distinguish between the aggregation of multiple species in the same region, but also determine whether it was more affected by the promotion or the environment. Therefore, researchers should pay attention to the influence of abiotic factors (such as habitat heterogeneity) in their future research and exploration of the spatial pattern of species. We also found that the promotion between trees with different sizes or different life history strategies was very weak, which may be due to the shelter and concealment of small trees or seedlings by large trees or shrubs and the reduction of the damage to young trees by external adverse conditions [5,50]. In addition, the interaction between species is a

dynamic process of change affected by the scale of study [5,7], which may be promoted at a certain scale, but as the scale changes, it may become competitive or not relevant [51]. For example, the small demand of individuals with a small diameter level for nutrients, living space, and other resources cannot affect the normal growth of individuals with a large diameter level, and individuals with a large diameter level may in turn promote the growth of individuals with a small diameter level by decomposing nutrients in the soil and resisting external habitat stress on a small scale. Individuals with a large diameter usually have highly developed stem and root systems, which make it easier for them to obtain and utilize resources. With an increase in the scale, the interspecific competition at the root tips of individuals with a large diameter becomes more intense. When the habitat stress is enough to affect the normal survival of individual species, adjacent species can be transformed into mutual promotion to resist external interference. For local tree species, this may be a mutually reinforcing way of survival in order to adapt to the high-altitude habitats, where *Carpinus tientaiensis* grow. Therefore, the interaction between species in the community is complex and changeable [40,45,49], and it has a unique research value and significance. During the natural succession of plant communities, the mechanisms of habitat filtration, biological competition, and promotion may drive the process of community construction [52,53].

### 4.3. Suggestions for the Protection and Propagation of Carpinus tientaiensis

Zhejiang Jingning has discovered the largest *Carpinus tientaiensis* community to date. A larger population has a higher value and significance in terms of studying how to protect the rare and endangered species, *Carpinus tientaiensis*. While there are more seedlings in the *Carpinus tientaiensis* population, and the regeneration is good, the survival rate of seedlings is low, and the competition is weak. For the future protection and management of *Carpinus tientaiensis*, we suggest the following: (1) Cutting should be conducted during fixed periods of the *Carpinus tientaiensis* sample plots to properly reduce the local density of highly enriched species and thus reduce community closure and competition among different species, which would increase the survival rate of *Carpinus tientaiensis* adult individuals. For example, *Carpinus tientaiensis* likes light, so the shade of *Cyclobalanopsis multinervis* and *Rhododendron fortunei* should be reduced in order to promote the growth and development of *Carpinus tientaiensis*. More suitable habitat conditions should be created to enhance the competition of seedlings and small trees for light and nutrients and accelerate the growth of seedlings and small trees. (2) Appropriate fertilization or protection of *Carpinus tientaiensis* seedlings and small trees to enhance their competitiveness. (3) In future studies, the niche and adaptability of *Carpinus tientaiensis* populations will be increased, and we will analyze which tree species can promote the growth and development of *Carpinus tientaiensis* with reasonable planting. (4) We also suggested looking for habitat conditions in which the chance of the development of this species is greater. By exploring the optimum environment for the population growth of the *Carpinus tientaiensis*, the propagation is carried out to increase the population.

### 5. Conclusions

The *Carpinus tientaiensis* population and common species in the sample land occupy a similar living space and adopt different ecological strategies to maintain their coexistence. With an increase in their diameter, the distribution of species decreases gradually from aggregation to random distribution. The spatial correlation between *Carpinus tientaiensis* and other species shows that the living space of the *Carpinus tientaiensis* population is gradually affected by the competition of shade-tolerant tree species (*Rhododendron fortune*, *Cyclobalanopsis multinervis*, etc.). Therefore, this study shows that the survival of *Carpinus tientaiensis* is threatened by the spatial pattern and interspecific association of species. From the results of the spatial correlation of different diameter levels of the *Carpinus tientaiensis* population, the hypothesis that the change process of species correlation drives the construction process of the forest community is supported. Through the study of the structure

and spatial distribution pattern of the *Carpinus tientaiensis* population, the formation and maintenance of forest ecosystems, the stability and succession law of communities, and the ecological characteristics and renewal of populations have been clarified, and detailed theoretical information for the guidance of forest ecosystem restoration and diversity protection has been provided, which is of great significance.

**Author Contributions:** Z.W. and C.W. designed this study and improved the English language and grammatical editing. L.Y. wrote the first draft of manuscript and performed the data analysis. L.Y., B.J., Y.X. and T.L. did the field works. W.Y. and J.Z. gave guidance and methodological advice. All the coauthors contributed to the discussion, revision and improvement of the manuscript. All authors have read and agreed to the published version of the manuscript.

**Funding:** This research was financially supported by the Major Collaborative Project between Zhejiang Province and the Chinese Academy of Forestry (No. 2019SY08) and the Natural Science Foundation of Anhui Province (1808085MC59).

**Acknowledgments:** We would like to thank Weiwei Zhao at the Hangzhou Normal University for his assistance with the English language and grammatical editing of the manuscript. And support of all staff of Zhejiang Hangzhou Urban Forest Ecosystem Research Station.

**Conflicts of Interest:** The authors declare no conflict of interest.

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
