# Peer review of "Competition Restricts the Growth, Development, and Propagation of Carpinus tientaiensis: A Rare and Endangered Species in China"

_forests, doi:10.3390/f12040503_

Round 1
Reviewer 1 Report
General remark
The paper has to be changed. However, the paper in its present form has serious shortcomings and flaws that would have to be correct before the paper could be consider for the acceptance. I am not native speaker but some sentences were hardly to understand. English language should be correct. References are incomplete.
Detailed comments:
- Pg.1, Title and Abstract – It´s necessary to write complete Latin name of species, i.e. Carpinus tientaiensis W.C. Cheng
- Pg 1., Abstract; It should be rewritten due to unbalanced structure (backround, methods, results, conclusions). Brief methodology is practically missing – Are there any variants in the trial? What relates on?
- Pg.1-3, Introduction – The name of the mentioned species is obscure for readers. In the title is Carpinus tientaiensis but in the rest of text Carpinos lientaiensis. Is it the same species? It should be explain and unify in the whole manuscript.
- Pg. 2, (Liu et al. 2020; Freckleton & Watkinson 2001) – authors should be ordered in ascending sequence according the years.
- Pg.2, Competition is influences also by other factors than DBH, spatial distribution etc. No reference related to relative height position, crown size, age, management, light conditions were mentioned in Introduction and Discussion, too.
- Pg.2, (2) goal of the paper - …..association vary with size of tree ….What does it mean “size of tree” – diameter, height, crown size? – Specified it. The scientific hypothesis should be formulate.
- Pg. 3, Materials and methods; part 2.2, last paragraph – please correct. We divided it into four categories according to DBH.
- Pg.4, part 2.4 - ……calculation process (, 2004)? – Authors are missing.
- Pg.4, Result; part 3.1, next to the last row – Rhododendron fortune – use italic letters.
- Pg.5, ……Carpinos lientaiensis – use italic letter.
- Pg.5, Figure 1 - in y – axis missing unit of measure (tree.ha-1)?? Again, Figure 1 is for Carpinus tientaiensis? Reference in the above-mentioned text is for Carpinos lientaiensis.
- The same comment for Figure 2, 3, 4, 5.
- Pg. 8, Table 1 – column “abundance” missing unit of measure (tree.ha-1)?? Basal area in “m”??? The unit of measure for basal area is square meter per hectare (m2.ha).
- Pg.9, Discussion……Therefore, on smaller spatial scale, resource competition ………, but interaction between individuals depends on the diameter difference between individuals. Really? What about effects of tree species, relative height position, crown site, light conditions?
- Pg.10, ……light and soil nutrients (Long at al. plos one, Clark et al. 1998) – missing year.
- Pg.11, part 4.3. – correct ncreasing to increasing.
- Pg.11-12, Conclusion – the sentences …..During the natural succession of plant communities ……..(Funk et al. 2017; Weither et al. 2011) – it´s not results of own research. Move it to Discussion.
- Pg.12, References should be identical with those used in the text of manuscript. Most of items are missing.
Author Response
Dear editor and reviewers
Thank you for your constructive comments and helpful suggestions on our manuscript-forests-1157265! We have made our best efforts in revising the manuscript according to your concerns. And also, to meet the language requirements of the journal, we uploaded our manuscript to the English language proofreading services assigned by Forests for language and structural in-depth proofreading (Fig1:English-Editing-Certificate-28342). We appreciate your consideration for publication of our manuscript in Forests. The following are our responses to the comments of the referees.
Fig 1 English-Editing-Certificate-28342
With best regards,
Runguo Zang on behalf of all the coauthors.
Responses to comments of reviewer 1
Question 1:Pg.1, Title and Abstract – It´s necessary to write complete Latin name of species, i.e. Carpinus tientaiensis W.C. Cheng
Response 1: Thank you for your concerns. We have clarified the descriptions Latin name of species on the title and Abstract, and final format I followed the recommendations of the native English speaking editors.
Question 2:Pg 1., Abstract; It should be rewritten due to unbalanced structure (backround, methods, results, conclusions). Brief methodology is practically missing – Are there any variants in the trial? What relates on?
Response 2: According to your suggestions, we rewrote the abstract. See Page1.
Revised paragraphs:
Abstract: The protection and propagation of rare and endangered species are key to the preservation of their population development; however, due to the scarcity of individuals, the potential effects and status of rare and endangered species in the whole forest ecosystem are still poorly understood. Using data from a 60*140m forest dynamic monitoring sample of the Carpinus tientaiensis species in Zhejiang of Southeast China. We assessed the population distribution and diameter at breast height (DBH) structure of the Carpinus tientaiensis species, which was a rare and endangered species, as well as intra- and interspecific correlation with other species The results show that saplings (1 cm ≤ DBH < 5 cm) and juveniles (5 cm ≤ DBH < 10 cm) were more aggregated than larger individuals (DBH ≥ 20 cm) of Carpinus tientaiensis. The DBH size structure of all the trees shows an obvious inverted “J” distribution. With an increase in the DBH size category, the number of individuals gradually decreases. Due to the diffusion limitation, the spatial distribution patterns of all the tree individuals and roof geese in the sample land are increased at a small spatial scale, and as the spatial scale increases, the degree of aggregation decreases gradually. The relationship between different diameter stages of the population of Carpinus tientaiensis showed a consistent general trend. The spatial distribution of individuals with a large diameter on a small scale was significantly positively correlated(P<0.001). With an increase in the scale, there was no significant correlation between individuals with a large diameter and individuals with a small diameter. There was no significant correlation between the population of Carpinus tientaiensis and other species in the sample, and the strong unidirectional competition of other species in the sample can be seen by the competition index. We found that interspecific competition restricts the growth and expansion of Carpinus tientaiensis, and it has adopted different ecological strategies to coexist with a population of common tree species occupying a similar living space.
Question 3: Pg.1-3, Introduction – The name of the mentioned species is obscure for readers. In the title is Carpinus tientaiensis but in the rest of text Carpinos lientaiensis. Is it the same species? It should be explain and unify in the whole manuscript.
Response 3: Yes, your statements are true. We corrected and unified the species names, which was Carpinus tientaiensis.
Question 4: Pg. 2, (Liu et al. 2020; Freckleton & Watkinson 2001) – authors should be ordered in ascending sequence according the years.
Response 4: Yes, your statements are true. We changed the order for authors should be ordered in ascending sequence according the years. See Pg. 2, “(Freckleton & Watkinson 2001; Liu et al. 2020)”.
Question 5: Pg.2, Competition is influences also by other factors than DBH, spatial distribution etc. No reference related to relative height position, crown size, age, management, light conditions were mentioned in Introduction and Discussion, too.
Response 5: Thanks for your advice. We thought this was no relevant to the content of the study, so we deleted the sentence.
Question 6: Pg.2, (2) goal of the paper - …..association vary with size of tree ….What does it mean “size of tree” – diameter, height, crown size? – Specified it. The scientific hypothesis should be formulate.
Response 6: We had stricted with scientific hypothesis. We changed “size of tree” to “the DBH size of tree”. See Pg.2, (2).
Question 7:Pg. 3, Materials and methods; part 2.2, last paragraph – please correct. We divided it into four categories according to DBH.
Response 7:Yes, your statements are true. We have corrected as “We divided it into four categories according to DBH”.
Question 8:Pg.4, part 2.4 - ……calculation process (, 2004)? – Authors are missing.
Response 8: We added authors as “(Wiegand and Moloney, 2004)”.
Question 9:Pg.4, Result; part 3.1, next to the last row – Rhododendron fortune – use italic letters.
Response 9: We corrected the error and used italic letters. See Pg.4, Result; part 3.1-“Rhododendron fortune”.
Question 10: Pg.5, ……Carpinos lientaiensis – use italic letter.
Response 10: We corrected the error and used italic letters. See Pg.5-“Carpinos tientaiensis”.
Question 11: Pg.5, Figure 1 - in y – axis missing unit of measure (tree.ha-1)?? Again, Figure 1 is for Carpinus tientaiensis? Reference in the above-mentioned text is for Carpinos lientaiensis.
Response 11: We added unit of measure (tree.ha-1). See Pg.5, Figure 1. And changed each “Carpinos lientaiensis” as “Carpinos tientaiensis”.
Question 12: The same comment for Figure 2, 3, 4, 5.
Response 12: We made corrections for Figure 2, 3, 4, 5.
Question 13: Pg. 8, Table 1 – column “abundance” missing unit of measure (tree.ha-1)?? Basal area in “m”??? The unit of measure for basal area is square meter per hectare (m2.ha).
Response 13: Yes, your statements are true. We corrected all units of measurement in the table. See Pg. 8, Table 1
Question 14: Pg.9, Discussion……Therefore, on smaller spatial scale, resource competition ………, but interaction between individuals depends on the diameter difference between individuals. Really? What about effects of tree species, relative height position, crown site, light conditions?
Response 14: Thanks for your advice. We changed as “Therefore, on a smaller spatial scale, resource competition leads to more obvious intraspecies and interspecific relationships, but their diameter differences are one of the determinants of interactions between individuals.”. See Pg.9.
Question 15: Pg.10, ……light and soil nutrients (Long at al. plos one, Clark et al. 1998) – missing year.
Response 15:We added year as “(Clark et al., 1998; Long et al., 2013)”. See Pg.10.
Question 16: Pg.11, part 4.3. – correct ncreasing to increasing.
Response 16: We corrected the mistake as “increasing”. See Pg.11, part 4.3.
Question 17: Pg.11-12, Conclusion – the sentences …..During the natural succession of plant communities ……..(Funk et al. 2017; Weither et al. 2011) – it´s not results of own research. Move it to Discussion.
Response 17: Thanks for your advice. We moved it to Discussion. See Pg.10.
Question 18: Pg.12, References should be identical with those used in the text of manuscript. Most of items are missing.
Response 18: We corrected the reference error. See Pg.12.

Reviewer 2 Report
Competition restricts the growth, development and propaga-tion of the Carpinus tientaiensis about rare and endangered species in China - In my opinion, the manuscript could deserve more detailed examination, but it is written in incomprehensible language. For this reason, I recommend sending the manuscript for linguistic proofreading, and a precise description of the methods and results. Moreover, even the citations in the parentheses is missing, and also the line numbers are missing.
I did not analyze the discussion due to the low-quality language of the work. Only when the methodology and results are clear one can relate to the discussion. Some detailed comments below
Introduction
Strange citation form: inside the sentence, and not justified bby the sentence construction : "Intra- and inter-specific inter-actions(competition and facilitation) (Callaway, 2007; Wright et al., 2014) has a significant role in constructing community structures and maintaining stable community coexist-ence(Kubota, 2007; Veblen, 2007; Muhamed et al., 2013a; Liu et al., 2020)."
The paragraph relating to plant competition could be more consize.
"The National Wild Key Conservation Plant List (First Batch) lists Rooftop Goose Ear as II protected iucn assessed as extremely dangerous.."
Carpinos lientaiensis - I didnot found such species in Zhao et al., 2020
"as II protected iucn assessed as extremely dangerous" - I found other, similar species Carpinus tientaiensis but with cathegory of critically endangered
"Which is the largest population found at present, and it is helpful to broaden the protection and habitat adaptability of Carpinos lientaiensis" - habitat adaptability?
"Studying the spatial structure and competitiveness of Carpinos lientaiensis populations is of great academic significance and conservation value for the study of the classifi-cation and flora of Birch and endangered populations." - such study can rather help in species protection, but I wouldnt say that such study is of great academic significance
"We looked forward to solving the following questions:" Here shoud be aims and hypotheses, besides you can answer the question , not solve it
"and the frost-free period is about 196 d. The daily average temperature ?10? and the annual accumulated temperature is 2 858~5 The annual growth period of vegetation ?200~240 d." - this part shoud be rewritten for better reading
"Forest coverage rate 96.5, the main tree species for Rhododendron fortune, Symplocos coreana, Pinus taiwanen-sis .etc." - as above
Materials and methods
Study area - some information source should be given
Sample surveys and data collection
According to the classification criteria of diameter-level structure of canopy trees in subtropical and temperate zones (Zhang et al., 2010; Liu et al., 2020), and combined the characteristics of the distribution about the radial order of the population of Carpinos lien-taiensis. - rewrite please this sentense
2.3. Competition index
"this paper can effectively reflect"?
the formula explanation is not clear, what does competitive wood mean?
2.4. Point pattern analysis method and Null Model
The pair-correlation function - unclear explanation
About univariate analysis.. - this part of the paragraph is unclear, please rewrite
Results:
showed obvious inverted J distribution?
at the scale of 0-5 - do you mean radious?
The spatial distribution of large diameter individuals on small scale is significantly positive correlation.??
Author Response
Dear editor and reviewers
Thank you for your constructive comments and helpful suggestions on our manuscript-forests-1157265! We have made our best efforts in revising the manuscript according to your concerns. And also, to meet the language requirements of the journal, we uploaded our manuscript to the English language proofreading services assigned by Forests for language and structural in-depth proofreading (Fig1:English-Editing-Certificate-28342). We appreciate your consideration for publication of our manuscript in Forests. The following are our responses to the comments of the referees.
Fig 1 English-Editing-Certificate-28342
With best regards,
Zhigao Wang on behalf of all the coauthors.
Response to reviewer 2
Question 1: Introduction Strange citation form: inside the sentence, and not justified bby the sentence construction : "Intra- and inter-specific inter-actions(competition and facilitation) (Callaway, 2007; Wright et al., 2014) has a significant role in constructing community structures and maintaining stable community coexist-ence(Kubota, 2007; Veblen, 2007; Muhamed et al., 2013a; Liu et al., 2020)."
Response 1: We've done deep language processing and editing. The corrected result as “Intra- and interspecific interactions (competition and facilitation) have a significant role in constructing community structures and maintaining a stable community coexistence (Kubota, 2007; Veblen, 2007; Callaway, 2007; Muhamed et al., 2013a; Wright et al., 2014; Liu et al., 2020).”.
Question 2: The paragraph relating to plant competition could be more consize.
Response 2: Thanks for your advice. We reorganized the language and paragraphs.
Question 3: "The National Wild Key Conservation Plant List (First Batch) lists Rooftop Goose Ear as II protected iucn assessed as extremely dangerous.."
Response 3: Thanks for your advice. We changed as “It is listed on the Red List of critically endangered species of the International Union for the Conservation of Nature (IUCN, http://www.iucnredlist.org), and a similar species, Carpinus tientaiensis, as critically endangered [26]”.
Question 4: Carpinos lientaiensis - I didnot found such species in Zhao et al., 2020
Response 4: We changed each “Carpinos lientaiensis” as “Carpinos tientaiensis”.
Question 5: "as II protected iucn assessed as extremely dangerous" - I found other, similar species Carpinus tientaiensis but with cathegory of critically endangered
Response 5: Thanks for your advice. We changed as “It is listed on the Red List of critically endangered species of the International Union for the Conservation of Nature (IUCN, http://www.iucnredlist.org)”.
Question 6: "Which is the largest population found at present, and it is helpful to broaden the protection and habitat adaptability of Carpinos lientaiensis" - habitat adaptability?
Response 6: Thanks for your advice. We changed as “This is the largest population found at present, and it is helpful for broadening the protection of this population and understanding the habitat adaptability of Carpinus tientaiensis (Zhao et al., 2020).”.
Question 7: "Studying the spatial structure and competitiveness of Carpinos lientaiensis populations is of great academic significance and conservation value for the study of the classifi-cation and flora of Birch and endangered populations." - such study can rather help in species protection, but I wouldnt say that such study is of great academic significance
Response 7: Thanks for your advice. We changed as “Studying the spatial structure and competitiveness of Carpinus tientaiensis populations is helpful in species protection, as it aids in the classification of the flora of Birch and endangered populations.”.
Question 8: "We looked forward to solving the following questions:" Here shoud be aims and hypotheses, besides you can answer the question , not solve it
Response 8: Thanks for your advice. We changed as “We used the following questions to guide the analyses:”.
Question 9:"and the frost-free period is about 196 d. The daily average temperature ?10? and the annual accumulated temperature is 2 858~5 The annual growth period of vegetation ?200~240 d." - this part shoud be rewritten for better reading
Response 9: Thanks for your advice. We rewrote the survey area as “The highest altitude is 1689.1 m; the annual average rainfall is 1918 mm; the annual average temperature is 11.8°C; the coldest monthly average temperature is 2.1 °C; the hottest monthly average temperature is 21.5 °C; the average relative humidity is more than 80%; the annual sunshine time is 1717.6 hours; and the frost-free period is about 196 d. The daily average temperature is ≥10 °C, and the annual accumulated temperature is 2858–5157 °C. The annual growth period of vegetation is greater than 200–240 d.”.
Question 10: "Forest coverage rate 96.5, the main tree species for Rhododendron fortune, Symplocos coreana, Pinus taiwanen-sis .etc." - as above
Response 10: Thanks for your advice. We rewrote the survey area as “The forest coverage rate is 96.5%, and the main tree species are Rhododendron fortune, Symplocos coreana, Pinus taiwanensis.”.
Question 11: Materials and methods. Study area - some information source should be given.
Response 11: Thanks for your advice. We added sources of information as “(Zhao et al., 2020)”.
Question 12:Sample surveys and data collection. According to the classification criteria of diameter-level structure of canopy trees in subtropical and temperate zones (Zhang et al., 2010; Liu et al., 2020), and combined the characteristics of the distribution about the radial order of the population of Carpinos lien-taiensis. - rewrite please this sentence
Response 12: We rewrote the sentence as “According to the classification criteria of the diameter-level structure of canopy trees in subtropical and temperate zones (Zhang et al., 2010; Liu et al., 2020), the characteristics of the distribution about the radial order of the population of Carpinus tientaiensis were combined.”.
Question 13:2.3. Competition index. "this paper can effectively reflect"?
Response 13: Thanks for your advice. By calculating the competition index of different species, reflect the competitiveness of different species and provide support for the difference of species spatial distribution and correlation.
Question 14:the formula explanation is not clear, what does competitive wood mean?
Response 14: Thanks for your advice. We reinterpreted the formula, and changed “competitive wood” as “the individuals which competing against object wood”.
Question 15:2.4. Point pattern analysis method and Null Model. The pair-correlation function - unclear explanation
Response 15: Thanks for your advice. We rewrote these two paragraphs.
“The pair-correlation function g(r) was used to analyze the spatial distribution of individual and population levels. The pair-correlation function g(r) is derived from the K function, mainly using the circle to replace the circle in the K function, excluding the cumulative effect in the calculation process (Wiegand and Moloney, 2004). That is, the degree of the spatial aggregation of species is analyzed by the single variable g(r) function, and the spatial correlation between two species is analyzed by the double variable g(r) function.
Null model selection: Environmental heterogeneity affects species distribution at large scales, and competition among species may be asymmetric (Wiegand et al., 2007). Considering that the survival rate of the same species is different in different habitats, individuals are jointly affected by habitat association and diffusion limitation. Therefore, the heterogeneous Poisson process is used to simulate the single species and bivariate statistics. The Heterogeneous Poisson process (HP) is mainly used to simulate the relationship between the species density function and its habitat in order to exclude the zero hypothesis model of habitat heterogeneity on a large scale and predict the probability of the occurrence of a population individual in a region with environmental covariates as a function. λ(s) shows the relationship between individual density and habitat heterogeneity through the spatial heterogeneity function (Illian et al., 2008; Liu et al., 2020). In the simulation process, we first fixed the position of one tree species, randomized the spatial distribution position of another tree species by the Heterogeneous Poisson process, and analyzed the changes of the spatial correlation between the two tree species. Then, keeping the position of the latter tree species unchanged, the spatial distribution of the former tree species was randomized by the Heterogeneous Poisson process, and the changes of the spatial correlation between the two tree species were analyzed again. A confidence interval of 99% was obtained using 100 Monte Carlo simulations, and the maximum distance scale was 60 m (Wiegand and Moloney, 2004). Concerning the univariate analysis, if the observed value was above the confidence interval, this indicated aggregation; if the observed value was below the confidence interval, this indicated a random distribution; and if the observed value was between the confidence intervals, this indicated a regular/hyperdispersed pattern. For bivariate analysis, if the observed value was above the confidence interval, this indicated a positive correlation; if the observed value was below the confidence interval, this indicated a negative correlation; and if the observed value was between the confidence intervals, this indicated that there is no significant correlation between the two species (Wiegand and Moloney, 2004; 2014). All the analyses were conducted using the “spatstat” package in R3.2.4 (R Core Team, 2016).”.
Question 16:About univariate analysis.. - this part of the paragraph is unclear, please rewrite Results:
Response 16: Thanks for your advice. We rewrote Results. See Pg.5-8.
Question 17: showed obvious inverted J distribution?
Response 17: Thanks for your advice. We changed as “The DBH size structure of all trees and Carpinus tientaiensis in the sample land is similar, the diameter structure of each population showed an obvious inverted J distribution, and the number of individuals decreased with the increasing diameter size in the sample (Figure 1).”.
Question 18:at the scale of 0-5 - do you mean radious?
Response 18: Thanks for your advice. This sentence was refers to Liu et al., 2020, who wrote the same in his article
Question 19:The spatial distribution of large diameter individuals on small scale is significantly positive correlation.??
Response 19: Thanks for your advice. We changed as “The relationship between different diameter levels of the Carpinus tientaiensis population shows a consistent overall trend. There was a significantly positive correlation (P<0.001) between individuals with a large diameter and individuals with a small diameter on a small scale.”.
Round 2
Reviewer 1 Report
All comments were accepted and/or explained.
Only little recommendation should be done: To be unified in the manuscript, values of no significance (P > 0.05, 0.01, 0.001?) add in the Abstract.
Author Response
Dear editor and reviewers
Thank you for your constructive comments and helpful suggestions on our manuscript-forests-1157265! We carefully revised the manuscript according to the opinions of the experts. Besides, we edited and modified the language about the manuscript through Forests (Fig1:English-Editing-Certificate-28342). Apart from the academic requirements of the journal, if a language still improvement is needed, we would like to have our manuscript be proofread by the English language proofreading services assigned by Forests to meet the language requirement of the journal.
We look forward to the publication of our manuscript in Forests. The following are our responses to the comments of the referees.
Fig 1 English-Editing-Certificate-28342
With best regards,
Zhigao Wang on behalf of all the coauthors.
Responses to comments of reviewer 1
Question 1: Only little recommendation should be done: To be unified in the manuscript, values of no significance (P > 0.05, 0.01, 0.001?) add in the Abstract.
Response 1: Thank you for your concerns. In the Abstract, we add the values of no significance (P > 0.05, 0.01, 0.001?) in detail, and to be unified in the manuscript. See Pg. 1. We changed as “With an increase in the scale, there was no significant correlation(P>0.05) between individuals with a large diameter and individuals with a small diameter. There was no significant correlation(P>0.05) between the population of Carpinus tientaiensis and other species in the sample, and the strong unidirectional competition of other species in the sample can be seen by the competition index.”
Reviewer 2 Report
The authors have improved the MS which is now more understandable. Aims are now better presented. Nevertheless, it seems that the language has not been improved enough. The discussion needs a lot of improvement. I get the impression that the authors don't know what to focus on. Not all results need to be discussed, some may be combined. The discussion should be considerably shortened and conclusions corrected. Detailed comments are presented below.
Introduction
"The distribution pattern of plants will determine the range of species competition, and the result of competition depends on the size of adjacent individuals " - not always the size
"For example, species that require more light usually exhibit aggregation distribution at smaller individual diameter levels, and random distribution patterns as the diameter level increases, which due to increased competition for resource requirements." - please check this sentence if there are no errors, I feel that "which" is here not necessary
Aim4 - it cannot be your aim
Materials and methods - general comment : unify please, either you use spaces before units or not
"The highest altitude is 1689.1 m " m asl (above see level)
"d" should still be written as "days"
"The whole sample was divided into 21 20 × 20 m samples by the total station, and then all of the 20 × 20 m samples were divided into 16 small samples of 5 × 5 m." again this sentence should be rewritten, what is 21 20?
"The competition index is a good index with which to study competition among species." - please rewrite, this construction is weird, also “useful index” would sound better,
Results:
"obvious inverted J distribution," change to "characteristic reverse J distribution"
"With an increase in the diameter level" just DBH
"Intraspecies association" - intraspecific?
"Interspecies association of Carpinus tientaiensis with other species" - as above interspecific, and this mean that it is an association with other species, so "with other species" should be deleted
"Other species with a large diameter (adults) in the sample showed a significantly positive
correlation to m seedlings" what is m seadlings?
"This is why the Carpinus tientaiensis population was not competitive in the community, and the adaptability to the relationship between the whole habitat and organisms is weak, so the growth is blocked, and the population size is small." this is not a result and should be moved to discussion if still needed
Discussion
General comment - the introduction to the discussion should go back to aims of the study and describe in the context of the results.
"Among the Carpinus tientaiensis communities, the competitive ability of shade-tolerant tree species is relatively strong, while that of strong positive tree species is relatively weak." weird sentence
"While the promotion of different size grades is observed within and between Carpinus tientaiensis species, the weak promotion is not the subject of community construction due to the habitat or other conditions." why not? please justify
"Early studies have shown that species populations are clustered at a certain scale, both in tropical and temperate zones." which early studies?
"From the results of interspecific association analysis on different diameter levels of the Carpinus tientaiensis population, the spatial distribution of individuals with a small diameter level on a small scale is significantly positively correlated, and with an increase in the scale, there was no significant correlation between individuals of a large diameter and those of a small diameter." - this is one sentece! besides discussion should not repeat the results, but put them in the context
"a result of the fierce competition" better just “competition”
"As we speculated, other species in the Carpinus tientaiensis community tend to be highly dispersed on smaller scales with an increase in the diameter level, which may be driven by intraspecific competition, and this is also in line with the conclusion predicted by the classical Janzen–Connell hypothesis” - why intraspecific, not interspecific?, something may be in line with the hypothesis, not the conclusion predicted, we see no conclusion yet, besides, please describe what is in line with the hypothesis and why?
"This also indicates that the mortality of the population is dependent on the density," it is a truism
"Therefore, we found that competition intensity in the Carpinus tientaiensis population was very small [38]." so you found that competition is small, thus why you cite this publication?
"Most successional pioneer species (e.g. deciduous species) use resource acquisition strategies to achieve rapid growth and reproduction." - citation?
"We speculate that even if there is strong competition among species, the competing species can coexist as long as the species competitiveness or competition is weakened by some external factors [41]." again, you speculate and cite
"Especially in bad habitats" what is bad habitat?
General comment - this paragraph should be shortened and focus on one or two ideas
4.3. Suggestions for the protection and propagation of Carpinus tientaiensis - well, cutting in my opinion does not lead to the creation of a permanent basis for the protection of the species, but forces incidental actions. I suggest looking for habitat conditions in which the chance of the development of this species is greater.
conclusions - most of them is a repetition of discussion. I suggest to move suggestions for protections to this section, but shortened
Author Response
Dear editor and reviewers
Thank you for your constructive comments and helpful suggestions on our manuscript-forests-1157265! We carefully revised the manuscript according to the opinions of the experts. Besides, we edited and modified the language about the manuscript through Forests (Fig1:English-Editing-Certificate-28342). Apart from the academic requirements of the journal, if a language still improvement is needed, we would like to have our manuscript be proofread by the English language proofreading services assigned by Forests to meet the language requirement of the journal.
We look forward to the publication of our manuscript in Forests. The following are our responses to the comments of the referees.
Fig 1 English-Editing-Certificate-28342
With best regards,
Zhigao Wang on behalf of all the coauthors.
Responses to comments of reviewer 2
Question 1: "The distribution pattern of plants will determine the range of species competition, and the result of competition depends on the size of adjacent individuals " - not always the size
Response 1: Thank you for your concerns. We changed as "The distribution pattern of plants will determine the range of species competition, and the size of adjacent individuals is also one of the factors that determine the intensity of competition". See Pg. 1.
Question 2: "For example, species that require more light usually exhibit aggregation distribution at smaller individual diameter levels, and random distribution patterns as the diameter level increases, which due to increased competition for resource requirements." - please check this sentence if there are no errors, I feel that "which" is here not necessary
Response 2: Thank you for your concerns. We checked the sentence and adopted your opinion. "For example, light-preferring species usually exhibit aggregation distribution at small diameter, while showing random distribution at larger diameter, which might due to increased resource competition."
Question 3: Aim4 - it cannot be your aim
Response 3: Thank you for your concerns. We accepted your opinion and deleted the sentence. See Pg. 3.
Question 4: Materials and methods - general comment : unify please, either you use spaces before units or not
Response 4: Thank you for your concerns. We accepted your opinion and unified the general comment and so on. See Pg. 3.
Question 5: "The highest altitude is 1689.1 m " m asl (above see level)
Response 5: Thank you for your concerns. We changed as “The highest altitude is 1689.1m asl (above sea level).” See Pg. 3.
Question 6: "d" should still be written as "days"
Response 6: Thank you for your concerns. We changed as “day”. See Pg. 3.
Question 7: "The whole sample was divided into 21 20 × 20 m samples by the total station, and then all of the 20 × 20 m samples were divided into 16 small samples of 5 × 5 m." again this sentence should be rewritten, what is 21 20?
Response 7: Thank you for your concerns. We changed as “The whole plot was divided into 21 20 × 20m samples by the total station, and then all of the 20 × 20 m samples were divided into 16 small quadrats of 5 × 5m.”. See Pg. 3.
Question 8: "The competition index is a good index with which to study competition among species." - please rewrite, this construction is weird, also “useful index” would sound better,
Response 8: Thank you for your concerns. We changed as “The competition index is a useful index with which to study competition among species.” See Pg. 3.
Question 9: "obvious inverted J distribution," change to "characteristic reverse J distribution"
Response 8: Thank you for your concerns. We changed as “characteristic reverse J distribution,”. See Pg. 5.
Question 9: "With an increase in the diameter level" just DBH
Response 9: Thank you for your concerns. We changed as “With the increase of DBH,”. See Pg. 5.
Question 10: "Intraspecies association" - intraspecific?
Response 10: Thank you for your concerns. We changed as “Intraspecific association”. See Pg. 7.
Question 11: "Interspecies association of Carpinus tientaiensis with other species" - as above interspecific, and this mean that it is an association with other species, so "with other species" should be deleted
Response 11: Thank you for your concerns. We changed as “Interspecific association of Carpinus tientaiensis population”. See Pg. 7.
Question 12: "Other species with a large diameter (adults) in the sample showed a significantly positive correlation to m seedlings" what is m seadlings?
Response 12: Thank you for your concerns. We changed as “Other species with a large diameter (adults) in the sample showed a significantly positive correlation to seedlings”. See Pg. 7.
Question 13: "This is why the Carpinus tientaiensis population was not competitive in the community, and the adaptability to the relationship between the whole habitat and organisms is weak, so the growth is blocked, and the population size is small." this is not a result and should be moved to discussion if still needed
Response 13: Thank you for your concerns. We accepted your suggestion and changed this sentence to the discussion section. See Pg. 10.
This one-way competition is mainly the competition of the individual tree of the canopy for the light and resources of the small individual under the forest. This is why the Carpinus tientaiensis population was not competitive in the community, and the adaptability to the relationship between the whole habitat and organisms is weak, so the growth is blocked, and the population size is small. As a result, we speculate that asymmetric interspecific competition plays a relatively strong role in shaping the individual structure and species coexistence of the Carpinus tientaiensis populations in this forest community.
Question 14: "Among the Carpinus tientaiensis communities, the competitive ability of shade-tolerant tree species is relatively strong, while that of strong positive tree species is relatively weak." weird sentence
Response 14: Thank you for your concerns. We changed as “Among the Carpinus tientaiensis communities, the competitive ability of shade-tolerant tree species is relatively strong, while that of strong light-preferring tree species is relatively weak.” See Pg. 9.
Question 15: "While the promotion of different size grades is observed within and between Carpinus tientaiensis species, the weak promotion is not the subject of community construction due to the habitat or other conditions." why not? please justify
Response 15: Thank you for your concerns. We changed as “While the weaker promotive effect of different size grades is observed within and between Carpinus tientaiensis species. However, the frequency and intensity of promotion was much lower than that of competition due to habitat or other conditions. Therefore, weak promotion may not be the main driving force of community construction.” See Pg. 9.
Question 16: "Early studies have shown that species populations are clustered at a certain scale, both in tropical and temperate zones." which early studies?
Response 16: Thank you for your concerns. We added references to support this sentence. “Early studies have shown that species populations are clustered at a certain scale, both in tropical and temperate zones[32].” See Pg. 9.
- Araujo, M. S., and Costa-Pereira, R. 2013. Latitudinal gradients in intraspecific ecological diversity. Biology Letters. 9:20130778.
Question 17: "From the results of interspecific association analysis on different diameter levels of the Carpinus tientaiensis population, the spatial distribution of individuals with a small diameter level on a small scale is significantly positively correlated, and with an increase in the scale, there was no significant correlation between individuals of a large diameter and those of a small diameter." - this is one sentece! besides discussion should not repeat the results, but put them in the context
Response 17: Thank you for your concerns. We changed as “We found that the spatial distribution of individuals of the Carpinus tientaiensis population with a small diameter level on a small scale is significantly positively correlated, and with an increase in the scale, there was no significant correlation between individuals of a large diameter and those of a small diameter.” See Pg. 9.
Question 18: "a result of the fierce competition" better just “competition”
Response 18: Thank you for your concerns. We changed as “competition”. See Pg. 9.
Question 19: "As we speculated, other species in the Carpinus tientaiensis community tend to be highly dispersed on smaller scales with an increase in the diameter level, which may be driven by intraspecific competition, and this is also in line with the conclusion predicted by the classical Janzen–Connell hypothesis” - why intraspecific, not interspecific?, something may be in line with the hypothesis, not the conclusion predicted, we see no conclusion yet, besides, please describe what is in line with the hypothesis and why?
Response 19: Thank you for your concerns. We added references to support this sentence. "As we speculated, other species in the Carpinus tientaiensis community tend to be highly dispersed on smaller scales with an increase in the diameter level, which may be driven by intraspecific competition, and this is also in line with the conclusion predicted by the classical Janzen–Connell hypothesis [2,38].”
Question 20: "This also indicates that the mortality of the population is dependent on the density," it is a truism
Response 20: Thank you for your concerns. Yes, we thought this that this theory can accurately reveal the effect of density on individual mortality. See Pg. 9.
Question 21: "Therefore, we found that competition intensity in the Carpinus tientaiensis population was very small [39]." so you found that competition is small, thus why you cite this publication?
Response 21: Thank you for your concerns. We changed as “Therefore, we found that competition intensity in the Carpinus tientaiensis population was very small [25].” See Pg. 10.
Question 22: "Most successional pioneer species (e.g. deciduous species) use resource acquisition strategies to achieve rapid growth and reproduction." - citation?
Response 22: Thank you for your concerns. We added references to support this sentence. “Most successional pioneer species (e.g. deciduous species) use resource acquisition strategies to achieve rapid growth and reproduction[2].” See Pg. 10.
Question 23: "We speculate that even if there is strong competition among species, the competing species can coexist as long as the species competitiveness or competition is weakened by some external factors [42]." again, you speculate and cite
Response 23: Thank you for your concerns. We added references to support this sentence. “We speculate that even if there is strong competition among species, the competing species can coexist as long as the species competitiveness or competition is weakened by some external factors [4,17,43].” See Pg. 10.
Question 24: "Especially in bad habitats" what is bad habitat?
Response 24: Thank you for your concerns. We changed as “Especially in habitat with poor conditions such as low temperature, drought or barrenness” See Pg. 10.
Question 25: 4.3. Suggestions for the protection and propagation of Carpinus tientaiensis - well, cutting in my opinion does not lead to the creation of a permanent basis for the protection of the species, but forces incidental actions. I suggest looking for habitat conditions in which the chance of the development of this species is greater.
Response 25: Thank you for your concerns. Yes, I fully agree with you about finding habitat conditions for this species to develop more opportunities. However, in the existing communities about Carpinus tientaiensis, we cut down other tree species to reduce the competition for the Carpinus tientaiensis population, which is also beneficial to the growth and development of the Carpinus tientaiensis, also can improve the survival rate. We also add the advice as “(4) We also suggested looking for habitat conditions in which the chance of the development of this species is greater. By exploring the optimum environment for the population growth of the Carpinus tientaiensis, the propagation is carried out to increase the population.” See Pg. 12.